

# OWO
## Organized Waste Optimization



**Autors**: Bartosz Szewczyk · Nikodem Stamirski · Zuzanna Sztyma

**Supervisor:** Natalia Piórkowska

### Abstract

The "Organized Waste Optimization" (OWO) project aims to improve waste management in Wrocław through an innovative distributed system. This system integrates IoT-enabled smart bins, a centralized back-end for data processing and route optimization, and a front-end application for visualization and tracking. Key benefits include reduced fuel consumption, minimized traffic congestion, and a cleaner urban environment. Despite risks such as environmental challenges and scalability, our prototype highlights the potential for practical implementation.

## 1 DEVELOPMENT

### 1.1 Introduction

Effective waste management is essential in modern cities, yet inefficiencies persist due to static collection schedules and overflowing bins. This project aims to address these issues by developing a distributed smart bin system that dynamically monitors waste levels and generates optimized routes for garbage collection.

**Objectives**

- Create IoT-enabled smart bins capable of monitoring waste levels and environmental conditions.

- Develop a robust back-end for real-time data aggregation, storage, and optimization.

- Design a front-end application for route visualization and system management.

### 1.2 Related Work

In Wrocław and most other cities, a fixed route solution is usually implemented and calculated using Vehicle Routing Problem (VRP) optimization algorithms but the input data for these algorithms is often based on city demographics which might not always be the most efficient.

Other than our project, there are other existing solutions which try to integrate existing bins into smart optimizations system:

**Sensoneo:** Their bins include various additional but non-essential sensors. Their single rangefinder is relatively large yet offers only a 2.5-meter range. Additionally, they use a one-time battery instead of a solar panel.

**Enevo:** Enevo's bins also feature non-essential sensors, such as a thermometer. They rely on less efficient LTE-M connectivity (with a shorter range than LoRa), but they provide a fully automated service using ML models. Enevo also offers different bin models for various use cases.

### 1.3 Results

#### 1.3.1 Smart Bin System

Microcontroller used for this project is ESP32 on the Arduino Nano ESP32 board.

**Communication with the server**   The Grove-Wio-E5 RF module enables communication with the server using LoRa and LoRaWAN technologies. LoRa modulation allows the device to transmit data over distances of up to 10 km under ideal conditions. With its long range and low power consumption, LoRa is well suited for transmitting simple data, such as fill levels or location information, to a server.

   The LoRaWAN protocol is built on top of the LoRa modulation and is used to manage communication between devices in a network. LoRaWAN is a Media Access Control (MAC) layer protocol that defines how LoRa devices interact with the network, enabling them to join the network, authenticate, and exchange data.

### Sending data to LoRaWAN network works as follows:

1. Microcontroller communicates with the Grove-Wio-E5 using UART. To facilitate this communication, the device relies on AT commands.

2. Grove-Wio-E5 transmits data using LoRa modulation and the LoRaWAN protocol.

3. The LoRaWan gateway receives LoRa signals from the end devices.

4. The LoRaWan gateway forwards the received data to The Things Network (TTN) server for further processing.

**Analysis of the fill level**   of the container is performed using 4 ultrasonic range sensors HC-SR04. The sensors are positioned at the top of each wall of the bin, facing downward to measure the distance from the objects in the container.

   Activating all 4 sensors simultaneously would lead to signal interference because the sensors emit ultrasonic waves at the same frequency. These overlapping signals make it difficult for the sensors to distinguish their own reflected waves. Therefore, sensors must be activated sequentially.

### Steps for calculating approximate fill level:

1. A single sensor makes 10 measurements.

2. Outliers are identified and removed using the standard deviation method.

3. The remaining data is averaged to produce a single reliable distance measurement.

4. The process is repeated for the rest of the sensors.

5. The percentage of fill level for each sensor is calculated based on the height of the container.

6. The final average fill level of the container is determined by averaging the result from all 4 sensors.

**Power supply and energy management**   is handled by a Waveshare Solar Panel (18V, 10W), which works with a solar power manager to charge a 3.7V lithium-ion battery with a capacity of 10,000mAh.

- The battery provides enough energy to keep the bin system running in sleep mode for up to 69 hours, ensuring continuous operation at night when sunlight is unavailable.

- In perfect condition, the solar panel produces 10W of power and, accounting for 90% efficiency, generates approximately 9W. With the bin system consuming 0.48W the solar panel alone can supply enough energy to power the entire system.

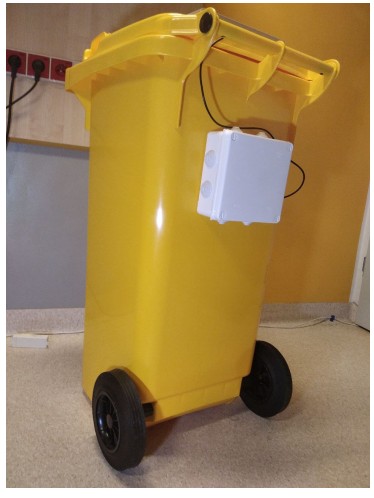

(a) Distribution box with micro-controller, voltage regulator with battery, GPS and LoRa modules

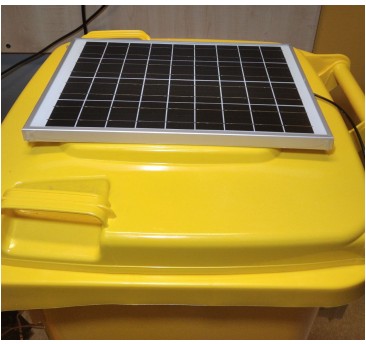

(b) Solar panel

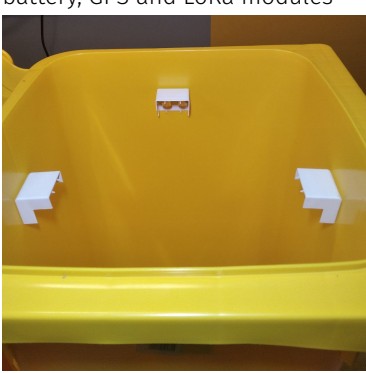

(c) Range sensors inside the bin box

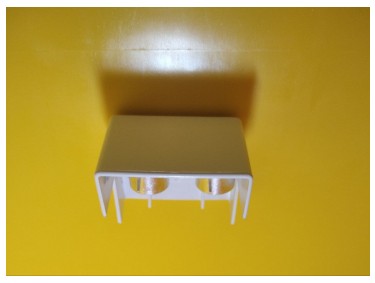

(d) Range sensor

### 1.3.2 LoRaWAN Connectivity: The Things Network

**The Things Network** is a service for LoRaWAN connectivity that we used in this project. Its main functions are:

- Registering an application.

- Registering LoRa devices in an application.

- Receiving data from LoRa devices through LoRa gateways.

- Storing this data and providing multiple APIs to process it by other applications.

- Providing an additional API for project manipulation outside its own web interface.

We use this service with our own application to insert and test a bin system and the back-end functionalities associated with this part of our project.

### 1.3.3 Back-end: Centralized Data Processing

**Overview** The back-end, written in Python using **Domain-Driven Design** and containerized, serves as the system's core, aggregating data from IoT devices, storing it in a PostgreSQL database, and generating optimized collection routes.

**Technologies and Implementation**

- **HTTP Framework** - Python-based FastAPI library for efficient data handling.

- **MQTT Framework** - Paho-MQTT library for receiving data from the bins through TTN.

- **Database** - PostgreSQL for structured data storage and retrieval.

- **Routing cost calculator** - OpenStreetMap (OSM) library was used for calculating costs between nodes which is needed for the route generation. The library contains a set of city maps with nodes and a built-in way of calculating different types of costs between nodes.

- **Quick cost lookup with cost maps** - An original cost map solution was created to store pre-calculated costs between nodes on a map to ensure quick route calculations as the OSM algorithm was slow.

- **Optimization Algorithm** - OR-Tools library and its set of functions for solving VRP problems calculate fuel-efficient and time-saving routes for garbage trucks.

- **Multi-threaded design** - Many parts of the system like MQTT connectivity, cost map updates, generating routes or HTTP interface run on separate threads to prevent it from bottle-necking.

**Architecture Diagram** showcases how the different layers of the back-end application are interconnected. Synchronization of objects between the Domain and Database layers has been done using the Repository Design Pattern.

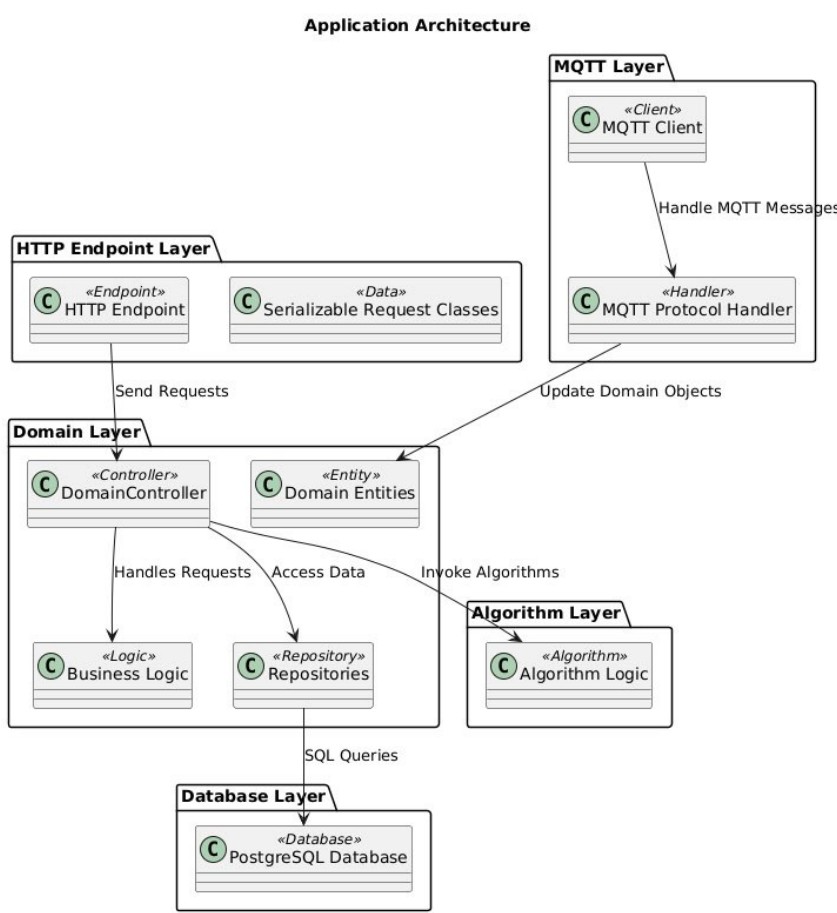

Figure 2: Back-end Architecture Diagram

**Cost Maps** are used to store pre-calculated costs of traveling between nodes to improve the optimization's speed. This solution also lets the costs for a single bin be easily updated with location reducing the complexity from $O(n^2)$ to just $O(n)$.

Figure 3: Cost Map Initialization

**Optimization Algorithm** created with the OR-Tools library's VRP solving functions is able to return a series of paths for all given trucks alongside information of total load gathered and total time used in carrying out the routes which are helpful in testing the used optimization options.

```
python-app  | Truck 1:
python-app  |   Route: [396, 481, 36, 1041, 401, 676, 376, 306, 841, 956, 776, 41
python-app  |   Load: 150/150
python-app  |   Distance: 529
python-app  | Truck 2:
python-app  |   Route: [461, 746, 701, 1201, 1036, 526, 1111, 71, 546, 1066, 186,
 236, 826]
python-app  |   Load: 150/150
python-app  |   Distance: 644
python-app  | Truck 3:
python-app  |   Route: [101, 11, 731, 771, 91, 131, 561, 551, 736, 196, 726, 16, ‹
6, 206]
python-app  |   Load: 150/150
python-app  |   Distance: 973
python-app  | Truck 4:
python-app  |   Route: [1196, 881, 286, 281, 276, 876, 861, 76, 221, 866, 871, 22(
python-app  |   Load: 150/150
python-app  |   Distance: 691
python-app  | Truck 5:
python-app  |   Route: [146, 346, 336, 331, 326, 1176, 1046, 791, 786, 356, 46, 5(
, 1106, 531, 1121]
python-app  |   Load: 150/150
python-app  |   Distance: 853
python-app  |
python-app  | Total solution cost: 3690
python-app  |
```

Figure 4: Optimization Algorithm Result

### 1.3.4 Front-end: User Interface and Visualization

**Overview** The front-end application provides interface for easy and efficient data access and management for waste management companies, ensuring efficient coordination and decision-making. It allows for access to all the projects features, like bin fill levels and calculated routes. It also provides map views showing locations of various bins and calculated routes. It uses a highly adaptive modular design allowing for reusability of components and easy maintenance.

**Functionalities Implemented** The front-end application of the OWO project includes several key functionalities:

· **Homepage:** A representative page listing the biggest selling points of the project in a modern style

· **Bin, Truck and Dispatch Center Administration screens:** Sections for managing smart bins, trucks and Dispatch Centers, including adding, editing, deleting and displaying data.

· **Dispatch Center Details:** A detailed view of a specific dispatch center data, including its dispatches and the ability to create new dispatches with a dropdown menu for garbage type selection

· **Map view:** A map view that displays the locations of bins and dispatch centers. It includes a dropdown to filter bins and centers by dispatch center and a popup with details when an object is clicked.

· **Dispatch details:** A view that displays the calculated paths for a specific disaptch on the map and lists all related data

· **Navigation view:** A map view showing the most recent path for a specified truck with real time location and path to the first assigned bin, allowing workers to follow the route easily

**Technologies used**

· **React:** A JavaScript library for building user interfaces, enabling the creation of reusable components.

· **TypeScript:** A typed superset of JavaScript that enhances code quality and maintainability.

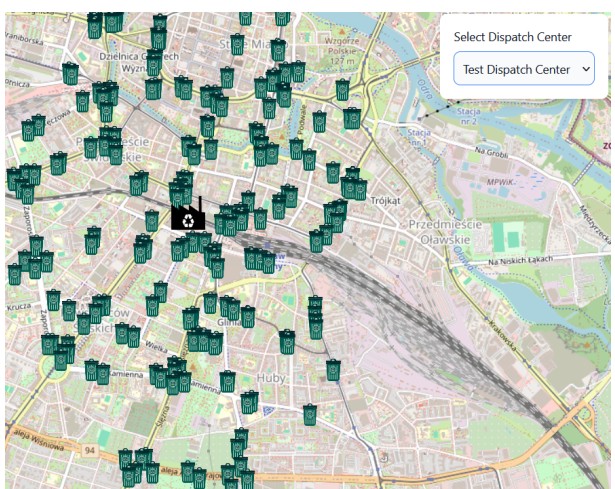

Figure 5: Map view

- **Vite:** A fast build tool that provides a modern development environment with hot module replacement (HMR).

- **Tailwind CSS:** A utility-first CSS framework that allows for rapid UI development with a consistent design system.

- **React Router:** A library for handling client-side routing, enabling seamless navigation between different views.

- **Leaflet:** An open-source JavaScript library for interactive maps, used to visualize bin locations and optimized routes.

- **React-Leaflet:** A React wrapper for Leaflet, making it easier to integrate maps into React components.

## 2 CONCLUSION

### 2.1 Conclusions

#### 2.1.1 Key Achievements

- Demonstrated the potential for reduced fuel consumption, cleaner streets, and lower workforce requirements in theoretical analyses.

- Successfully designed and implemented a back-end system that processes real-time data from smart bins.

- Optimized route generation using the OR-Tools VRP algorithm and a custom CostMap, achieving significant efficiency improvements compared to direct OSM-based calculations.

- Developed a scalable, extensible architecture based on Domain-Driven Design principles, enabling future growth and adaptability.

- Developed a user-friendly user interface allowing users to see, edit and delete data, and display it on maps

#### 2.1.2 Key Insights

- LoRaWAN technology is an excellent choice for IoT solutions in smart cities due to its low power consumption and long-range capabilities.

- Domain-Driven Design enables clear separation of concerns and makes the system easier to maintain and extend.

### 2.1.3  Limitations

- Lack of real-world implementation data limits the ability to validate the system's full effectiveness.

- The OR-Tools algorithm requires parameter tuning to accommodate real-world complexities like traffic patterns.

- Current hardware solution for bins is a prototype and needs a proper version.

## 2.2  Future Directions

### 2.2.1  Smart Bin System

1. Designing a proper mounting solution for bin computers inside of bins - Our solution is just a prototype, and a solution environment-resistant solution should be made.

2. Implementing Sleep Mode for ESP32 for power consumption optimization

3. Improving reliability of the measurements by implementing better outlier detection and noise reduction.

4. Improving the method of finding the fill level approximation by using specialized algorithms or machine learning models.

### 2.2.2  Back-End

1. Introduction of proper HTTP protocol security which is lacking in our solution - Without it deployment is not possible.

2. Introduction of more integrity checks and security in the domain and database - Data integrity between domain and database might be at risk in unexpected circumstances and a more robust security against SQL injection might be needed.

3. Incorporating an existing bin and truck identification solutions that are used by Wrocław for compatibility with other Wrocław's systems.

4. Playing more with the OR-tools VRP algorithm parameters or introducing new ones.

5. Introduction of an additional optimization algorithm that would calculate days for dispatches as the current solution requires a person to make that decision.

6. (Optional) Replacing the currently used VRP algorithm solution with a real-time ML model that would create dispatches when necessary and could predict filling.

7. (Optional) Storing different cost maps for different seasons or days or times of day or better cost predictions.

### 2.2.3  Front-End

1. Introduction of an authentication system for workers and administrators.

2. Enhancing user experience, especially in navigation mode.

3. Improving the visual aspect of the app.

4. Implementing real time updates from the server.

5. Adding advanced filtering and search to the administration screens.

6. Implementing input validation.

## 2.3  Acknowledgments

We want to give thanks to our amazing supervisor Natalia Piórkowska. Without her help and patience, this project wouldn't come together.

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
