# OpenReview forum: "OWO - Organized Waste Optimization"
_pwr.edu.pl/Wrocław_University_of_Science_and_Technology/2024/ZPI_Day — Wrocław University of Science and Technology 2024 ZPI Day Submission_

### Official Review · Reviewer_zerJ · 2024-12-05
**Organized Waste Optimization**

**Confidence:** 4
**Significance Of Results:** 4
**Overall Quality:** 3

**Compliance With Template:**

4: High Quality – The article contains all the required sections, which are well-written and substantively correct, although minor errors or shortcomings may be present. The overall structure is clear and coherent.

**Description Of Results:**

3: Average Quality – The results are described with moderate detail. Some examples or evaluation elements are present but insufficiently developed or incomplete.

**Feedback On Consistency:**

The document is well-structured and logically coherent, starting with an abstract and introduction that clearly define the problem and objectives, followed by detailed sections on development and results, and concluding with comprehensive conclusions and future directions. Each section effectively builds upon the previous one, guiding the reader through the project’s progression from problem identification to solution implementation and reflective analysis. To enhance the document’s consistency and depth, incorporating more quantitative data and metrics in the results section would provide tangible evidence of the system’s effectiveness and strengthen the connection between the problem analysis and the outcomes. Additionally, making explicit connections between specific results and how they address the initial problems would improve clarity and persuasive power, as some links are currently not sufficiently clear. Expanding on the limitations with specific examples or scenarios would also offer a deeper understanding of the challenges faced and demonstrate how future work plans to address them, thereby providing a more comprehensive and substantiated narrative.

**Potential For Development:**

The authors proposed natural areas for further development and application of the created solution.

**Project Nature Evaluation:**

The project addresses a key issue faced by modern cities: efficient waste management. Through dynamic monitoring of container fill levels and optimization of collection routes, the system can significantly improve operational efficiency. The authors have demonstrated practical application of their engineering skills to solve a real-world problem, which is commendable. The use of the ESP32 microcontroller, LoRa/WiFi modules, and HC-SR04 ultrasonic sensors highlights the application of modern IoT technologies in a practical context.

However, certain issues must be noted: the project is primarily based on a prototype and theoretical analyses, and the lack of conducted tests weakens the quality of the work. The OR-Tools algorithm requires further parameter adjustments to account for specific urban conditions, such as variable traffic patterns. Without proper tuning, the algorithm's efficiency may be limited. The authors completely overlooked this issue and did not propose any solutions to address it.

Although the system is powered by solar panels and batteries, long-term energy independence could be compromised under conditions of insufficient sunlight or increased energy demand, which is a common problem due to the frequency of waste peaks. The authors appear not to have conducted a risk analysis in this regard.

Another noteworthy aspect is the lack of advanced security mechanisms in both the back-end and front-end layers, which could expose the system to various threats. In an era where frameworks and tools supporting authorization and authentication processes are readily available, the absence of their implementation seems entirely unjustifiable, posing a significant problem in an engineering context.

**Technical Language Precision:**

5: Very High Quality – The language is entirely appropriate for a technical report. All terms are used correctly and precisely, and the style is professional, clear, and coherent, without any errors or ambiguities.

---

### Official Review · Reviewer_jaMP · 2024-12-06
**Kosmiczne rozwiązanie na starożytny problem :)**

**Confidence:** 4
**Significance Of Results:** 3
**Overall Quality:** 5

**Compliance With Template:**

5: Very High Quality – The article contains all the required sections, which are written in a very detailed, clear, and error-free manner. The structure is professional and meets expectations, and the content adheres to the highest substantive and formal standards.

**Description Of Results:**

5: Very High Quality – The results are described in detail, clearly and comprehensively, supported by thorough evaluation, analysis, and convincing usage examples. The description meets the highest substantive standards.

**Feedback On Consistency:**

Praca spójna i logiczna

**Potential For Development:**

Bez wsparcia firmy, która chciałaby to wdrożyć choćby na małą skalę, pilotażowo - nie uda się zmierzyć wydajności/oszczędności, czyli sensowności projektu.

Firmy wywozowe puszczają regularnie swoje śmieciarki stałymi trasami, gdy już wszystkie pojemniki są przepełnione, bo jeżdżenie częściej nawet zoptymalizowanymi trasami jest po prostu nieopłacalne. Poza tym istnieje kalendarz wywozów, jak to się ma do pomysłu?

**Project Nature Evaluation:**

Projekt typu proof-of-concept raczej bez możliwości rzeczywistego wdrożenia.
Literatura nie jest cytowana, ale i tak dotyczy tylko aspektów softwareowych. Brak analizy rynku i rzeczywistych potrzeb.
Ze zdjęć pojemnika widać niedojrzałość prototypu: panel słoneczny roztrzaskałby się przy pierwszym otwarciu pokrywy o skrzynkę z elektroniką. Czujniki zostałyby zerwane przy pierwszej próbie upchnięcia w pojemniku kartonu (tak, wiem... żółty nie jest do papieru :) ... no to "steropianu po odpakowaniu TV" :) .

**Technical Language Precision:**

5: Very High Quality – The language is entirely appropriate for a technical report. All terms are used correctly and precisely, and the style is professional, clear, and coherent, without any errors or ambiguities.

---

### Official Review · Reviewer_BxPw · 2024-12-08
**IT project prepared correctly.**

**Confidence:** 5
**Significance Of Results:** 5
**Overall Quality:** 5

**Compliance With Template:**

5: Very High Quality – The article contains all the required sections, which are written in a very detailed, clear, and error-free manner. The structure is professional and meets expectations, and the content adheres to the highest substantive and formal standards.

**Description Of Results:**

5: Very High Quality – The results are described in detail, clearly and comprehensively, supported by thorough evaluation, analysis, and convincing usage examples. The description meets the highest substantive standards.

**Feedback On Consistency:**

The analysis of the problem, presentation of results and conclusions are coherent and logical

**Potential For Development:**

The article indicates possibilities for further work or practical application of its results.

**Project Nature Evaluation:**

Both the level of usability, the applied technical methods and technological solutions have the characteristics of engineering work.

**Technical Language Precision:**

5: Very High Quality – The language is entirely appropriate for a technical report. All terms are used correctly and precisely, and the style is professional, clear, and coherent, without any errors or ambiguities.

---

### Decision · Program_Chairs · 2024-12-10

Accept (Oral)